# KRLS: Improving End-to-End Response Generation in Task Oriented Dialog with Reinforced Keywords Learning

**Xiao Yu, Qingyang Wu, Kun Qian, Zhou Yu**

Columbia University

`{xy2437,qw2345,kq2157,zy2416}@columbia.edu`

## Abstract

In task-oriented dialogs (TOD), reinforcement learning (RL) algorithms train a model to directly optimize response for task-related metrics. However, RL needs to perform exploration, which can be time-consuming due to the slow auto-regressive sequence generation process. We investigate an approach to create a more efficient RL-based algorithm to improve TOD performance in an offline setting. First, we use a faster generation procedure that samples from independent next-word distributions after training the language model (LM) with supervised learning. We then introduce a fine-grained reward function to help the model focus on learning key information in a dialog, by measuring the importance and semantic closeness of each generated token. Experiments on the MultiWoZ dataset show our new training algorithm, **K**eywords **R**einforcement **L**earning with Next-word **S**ampling (KRLS), achieves state-of-the-art performance on the end-to-end response generation task, with a 15% training time reduction compared to a standard RL algorithm using auto-regressive generation[1].

## 1 Introduction

Task-oriented dialog systems help users complete pre-defined tasks such as booking a hotel or reserving a table in a restaurant. With advances in large-scale pre-trained generative models (Brown et al., 2020; Raffel et al., 2020; Zhang et al., 2020c; Peng et al., 2022), many recent approaches (Wu et al., 2019; Hosseini-Asl et al., 2020; Yang et al., 2021a; Lee, 2021; He et al., 2022) handle TOD as a holistic task of end-to-end (E2E) generation, as opposed to the traditional modular approach.

In this E2E setting, the dialog history is often used as input, and reinforcement learning (RL) algorithms can train a model to generate a response that directly optimizes for task-related metrics, such as the task success rate (Budzianowski

---

[1]Code availale at https://github.com/jasonyux/KRLS

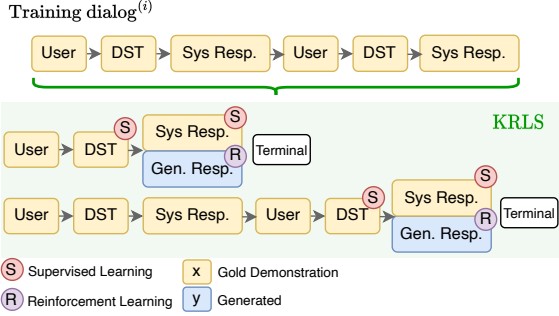

Figure 1: Overview of episodic training in KRLS. Each turn is treated as a separate episode, and the RL task is to only generate/explore a response at the end. The gold system resp. are also used in our reward computations.

et al., 2018). However, training RL-based dialog models often requires a good user simulator (Shi et al., 2019), and the training process could be time-consuming as RL often needs to explore and auto-regressively generate many new responses per given input (Ramamurthy et al., 2022). For example, in our experiment, we found that this generation process alone takes 172 minutes/epoch out of the total 362 minutes/epoch during training.

In this work, we aim to create a more efficient RL procedure for TOD, which does not need a user simulator nor use auto-regressive generation during exploration. We propose a new training algorithm, **K**eywords **R**einforcement **L**earning with Next-word **S**ampling (KRLS), which combines a faster sequence generation procedure and a fine-grained per-token reward in an offline setting (Jaques et al., 2019; Pang and He, 2021). First, we treat each turn in a dialog as a separate episode and consider an offline RL task to generate a response only at the end of each episode (see Figure 1). Since this procedure only explores/generates system responses at the last turn, no interactive environment (e.g., a user simulator) is needed. During this RL process, KRLS generates new sequences by directly sampling from independent next-word

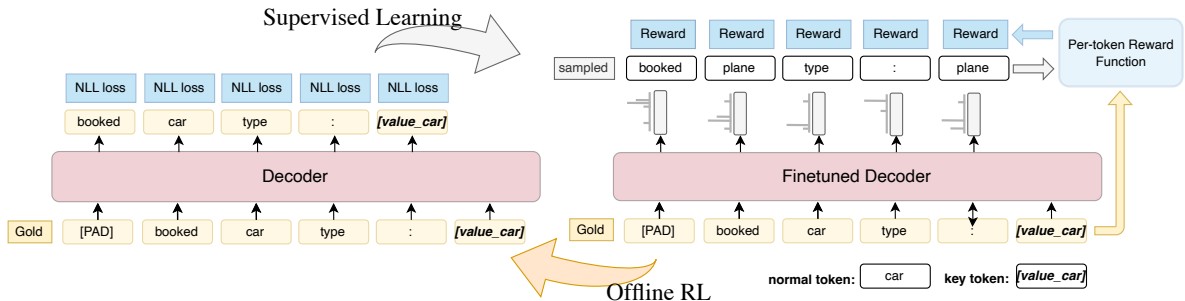

Figure 2: Overview of the KRLS algorithm. During traditional supervised training, the language model learns/imitates the gold response. During offline RL training, the fine-tuned model generates sequences by sampling from next-word distributions conditioned on the gold response and receives a per-token reward.

distributions, after training a language model with the traditional supervised learning (SL) technique (see Figure 2). This generation procedure is much faster than the traditional auto-regressive approach, as it only requires a single forward pass. Next, KRLS uses a fine-grained per-token reward to help the model focus on learning key information in a dialog, by measuring the importance and semantic closeness of each generated token. Experiments on the MultiWoZ dataset show that KRLS achieves state-of-the-art performance on the E2E response generation task, with a 15% training time reduction compared to the standard RL approach using auto-regressive generation.

This paper makes the following contributions:

- We propose an efficient offline RL algorithm that approximates auto-regressive generation by sampling from independent next-word distributions conditioned on the gold response.

- We introduce a per-token reward function, which can be used in our offline RL algorithm to promote keyword learning or to incorporate domain knowledge.

- We show that our proposed KRLS algorithm can achieve state-of-the-art performance on E2E response generation on MultiWoZ (Budzianowski et al., 2018; Eric et al., 2019; Zang et al., 2020).

## 2 Background

To introduce RL in NLP tasks, we begin by formulating the response generation process as an MDP. Given a supervised dataset $\mathcal{D} = \{(\mathbf{c}^i, \mathbf{x}^i)\}$ where

$\mathbf{c}^{(i)}$ is the context and $\mathbf{x}^{(i)}$ is a response, the probability of generating $\mathbf{x}^{(i)}$ can be modeled as:

$$p(\mathbf{x}^{(i)}|\mathbf{c}^{(i)}) = \prod_{t=1}^{T-1} p(x_{t+1}^{(i)}|x_{1:t}^{(i)}, \mathbf{c}^{(i)}),$$

where $x_t^{(i)}$ is the $t$-th token in the $i$-th response, and $T$ is the length of the response. As mentioned in Ramamurthy et al. (2022); Lubis et al. (2020), this generation can be formulated as a MDP problem $\langle \mathcal{S}, \mathcal{A}, \mathcal{R}, \mathcal{P}, \gamma \rangle$. The input context $\mathbf{c}^{(i)}$ would be the initial state $s_0 \in \mathcal{S}$, and the response $\mathbf{x}^{(i)}$ would represent the sequence of actions $\mathbf{a}^{(i)} = \{a_1^{(i)}, a_2^{(i)}, \ldots, a_{T-1}^{(i)}\}$ in an episode, where $a_t^{(i)} \in \mathcal{A}$ is the $t$-th token in the $i$-th response. The reward function $\mathcal{R}$ would represent the "utility" of each action contributing towards the overall performance, such as task success in TOD. Typically, this is modeled by using $\mathcal{R}(s, a) = 0$ for non-terminal states, and $\mathcal{R}(s_T, a)$ for terminal states which can be computed by combining scores such as task success and BLEU (Ramamurthy et al., 2022; Arora et al., 2022). The transition function $\mathcal{P} : \mathcal{S} \times \mathcal{A} \rightarrow \mathcal{S}$ would deterministically append the action $a_t$ to the current state $s_t$ so that $s_{t+1} = (c_0, \ldots, c_m, a_0, \ldots, a_t)$. Finally, $\gamma \in [0, 1)$ is the discount factor.

## 3 Approach

In the MulitiWoZ dataset (Budzianowski et al., 2018; Eric et al., 2019; Zang et al., 2020), we observe that key information, such as restaurant "phone number" and "address", needs to be *generated* correctly in a response to achieve a high task success/inform rate. However, traditional SL uses a negative log-likelihood loss, which asks the

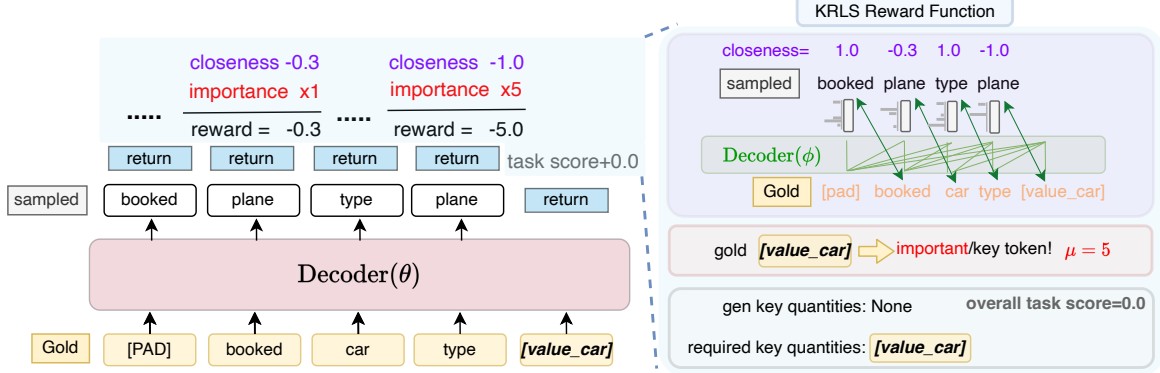

Figure 3: KRLS reward function. Immediate reward measures the semantic closeness of the generated token and the gold token, scaled by its importance $\mu$. Return for each sampled token is a combination of individual immediate reward and future rewards. Future rewards will help propagate final overall scores, such as overall task performance. Key tokens in a response are bolded and italicized.

language model to uniformly learn all correct tokens $x^{\mathrm{gold}}$ given input context $c$, without explicitly focusing on achieving task-related objectives:

$$\begin{aligned}\mathcal{L}_{\mathrm{SL}}(\theta) &= -\sum_x p(x|c)\log p_\theta(x|c) \\ &= -\log p_\theta(x^{\mathrm{gold}}|c)\end{aligned} \quad (1)$$

where the probability of $p(x|c) = 0$ if $x \neq x^{\mathrm{gold}}$. We will refer to models fine-tuned with this objective as "$\mathcal{L}_{\mathrm{SL}}$-finetuned".

We hypothesize that it can be beneficial to use RL and a fine-grained per-token reward function to help promote keyword learning and improve TOD performance. First, in Section 3.1 we propose a fast sequence generation procedure that can be used during RL training/exploration, and utilize policy-gradient based methods (Sutton et al., 1999; Williams, 1992; Schulman et al., 2017) to optimize response generation for task-related metrics. Then, in Section 3.2 we design a fine-grained reward function $\mathcal{R}(x|c)$ according to *how important* each generated token is, but also *how close* it is from the gold token, so that the model can focus more on learning key information once it can generate non-key tokens semantically close to the reference. Finally, we describe our KRLS algorithm in Section 3.3 and Algorithm 1, which utilizes RL combined with our proposed generation method and reward function.

### 3.1 RL with Next-word Sampling

To avoid the slow auto-regressive sequence generation process during RL training, we propose an alternative sequence generation mechanism that can be used for RL exploration. First, we assume

that there is a model $p_\theta$ that generates sequences similar to the gold responses in the *training set*. Then, under this assumption, we can approximate the MDP process of auto-regressive sequence generation by sampling from the next-word distributions conditioned on the gold response. This is because if previously generated tokens are similar to the gold context (e.g., after $\mathcal{L}_{\mathrm{SL}}$ training), then conditioning on those generated tokens is similar to conditioning on the gold tokens. In this setting, the next-word sampling process could approximate auto-regressive generation used during RL, but it is much faster as each token is generated in parallel.

Specifically, we first perform a forward pass to obtain the next-word distributions for $x_t^{\mathrm{gen}}$ given the context and the gold response up to $t-1$:

$$\left\{p_\theta(x_1^{\mathrm{gen}}|\mathbf{c}^{(i)}), \ldots, p_\theta(x_T^{\mathrm{gen}}|x_{1:T-1}^{(i)}, \mathbf{c}^{(i)})\right\} \quad (2)$$

Then, we generate each next-token $a_t = x_t^{\mathrm{gen}}$ by sampling from $p_\theta(X = x|x_{1:t-1}^{(i)}, \mathbf{c}^{(i)}; \tau)$ with temperature $\tau$. Finally, given some suitable reward function $\mathcal{R}(s_t, a_t) \in [-1, 1]$ (see Section 3.2 for details), we can use policy gradient methods (Sutton et al., 1999; Williams, 1992) to perform a "weighted learning" on each generated token:

$$\nabla\mathcal{L}(\theta) = -G_t \nabla\log p_\theta(x_t|c) \quad (3)$$

Note that this procedure is much faster than auto-regressive generation as it only requires one forward pass. Moreover, it is suitable for training a model to focus on generating key information, because we can use the gold response as an oracle to locate those key positions and penalize each generated token in the reward function accordingly.

## 3.2 Per-Token Reward Function

To improve a model's keyword generation ability, we design a reward function that measures the *importance* and *semantic closeness* of each generated token. This reward aims to prioritize accurate generation of key tokens, and also contextually evaluate how far-off is each generated token from the ground truth. We draw inspiration from BERTScore (Zhang et al., 2019), which uses a separate neural network to compute a contextual semantic score of the generated sequence by comparing it against the gold reference. However, we found that directly adapting BERTScore to a per-token reward function is sub-optimal in our setting, as our generated sequence is "sampled" from the gold response. Therefore, we consider a simpler approach, utilizing the fact that our generation procedure provided a one-to-one mapping between each generated token and the gold token.

First, we use a $\mathcal{L}_{\text{SL}}$-finetuned decoder network Decoder($\phi$) to compute the probability $p_\phi(X_t = x|x_{1:t-1}^{(i)}, \mathbf{c}^{(i)})$ of generating any token $x$ at time $t$, which can be done in a single forward pass. Then, we index into this probability distribution to find $p_\phi(X = x_t^{\text{gen}}|x_{1:t-1}^{(i)}, \mathbf{c}^{(i)})$ of our generated tokens, as a measure of the *semantic appropriateness* of $x_t^{\text{gen}}$ in the given context. To ensure that $\mathcal{R}$ correctly reflects the gold tokens as the optimal choice, we manually set this semantic closeness score to 1 for *any token* that is correctly generated $x_t^{\text{gen}} = x_t^{\text{gold}}$. To emphasize keyword learning, we also strictly set this closeness score to $-1$ if a *key token* is incorrectly generated. Otherwise, we use the probability $p_\phi$ produced by the decoder network as the closeness score. Finally, we adjust the reward for key tokens by an *importance* scale $\mu > 1$, which is a hyper-parameter for specifying the relative importance between keywords and non-keywords. This gives our per-token reward:

$$\mathcal{R}(s_t, a_t) = \text{closeness}(x_t^{\text{gen}}, x_t^{\text{gold}}, \bar{\mathbf{c}}) \cdot \mu \quad (4)$$

We standardize $\mathcal{R}$ to $[-1, 1]$ to be later compared with other related reward functions in Section 5.2.

## 3.3 KRLS Training Algorithm

Algorithm 1 describes our KRLS training algorithm. Given a supervised dataset $\mathcal{D} = \{\mathbf{c}_i, \mathbf{x}_i^{\text{gold}}\}_{i=1}^N$ consisting of the dialog context $\mathbf{c}_i$ and the gold response $\mathbf{x}_i^{\text{gold}}$ at each turn, we train a neural network $p_\theta$ to generate a response that satisfies the user's need given by $\mathbf{c}_i$. A separate neural

---

**Algorithm 1** KRLS Training Algorithm

---
**Require:** generative network $p_\theta$
**Require:** semantic scoring network $p_\phi$
**Require:** supervised language dataset $\mathcal{D}$
**Require:** empty buffer $B_{\text{L}}, B_{\text{S}}$
1: Repeat for $n$ epochs:
2:  **for** batch $b_i$ in $\mathcal{D} = \{b_1, \ldots, b_m\}$ **do**
3:     Perform sup. learning on $b_i$ (Equation 1)
4:     Update generative network $p_\theta$
5:     Append $b_i$ to buffer $B_{\text{L}}$
6:     **if** $i\% \kappa == 0$ **then**
7:         **for** each batched episode $b_j$ in $B_{\text{L}}$ **do**
8:             Collect $k$ samples per episode
9:               by sampling from Equation 2
10:             Calculate per-token reward
11:               using $p_\phi$ and Equation 4
12:             Calculate per-token returns $G_t$
13:             Append all to replay buffer $B_{\text{R}}$
14:         **end for**
15:         Perform RL on $B_{\text{R}}$ (e.g., PPO)
16:         Update generative network $p_\theta$
17:         Clear $B_{\text{L}}$ and $B_{\text{R}}$
18:     **end if**
19: **end for**

---

network $p_\phi$ is used to compute the reward function during RL phase (see Section 3.2).

For each epoch, we first perform SL over several batches of training examples and update our network $p_\theta$. This is because our generation procedure is based on the assumption stated in Section 3.1, so we periodically train $p_\theta$ with the $\mathcal{L}_{\text{SL}}$ objective to imitate the gold responses before passing over to RL (see Appendix F for generated sequences). Then, we store those SL-trained batches into a buffer $B_L$, and perform RL on this learned buffer. During this RL training, we first generate $k$ responses per trained episode by next-word sampling (see Section 3.1), calculate their rewards and returns using a scoring network $p_\phi$, and append them to a replay buffer $B_R$. Then, we utilize the clipped policy gradient objective from Proximal Policy Optimization (Schulman et al., 2017) to learn from $B_R$ and update the generative policy model $p_\theta$ (see Appendix A and Appendix B).

Note that Algorithm 1 only additionally requires prior definitions of keywords to compute the reward. This means that such an approach can be generic to many task-oriented dialogues where keywords can be easily defined (e.g., using entities from a database). For instance, in movie recom-

mendation (Harper and Konstan, 2015), correctly generating key information such as "movie_ratings" and "movie_genres" could be helpful to improve the system's response. Additionally, we believe that for other dialogue tasks such as QA/social chat, keywords can often be automatically processed and defined, such as using the entities mentioned in WikiQA answers (Yang et al., 2015) and the intent keywords in ESConv (Liu et al., 2021).

## 4 Experiments

### 4.1 Dataset and Preprocessing

We evaluate our algorithm on the MultiWoZ dataset (Budzianowski et al., 2018). MultiWoZ is a large-scale multi-domain TOD dataset consisting of 8438, 1000, and 1000 dialogs for training, validation, and test sets respectively. The dataset consists of seven domains: attraction, hotel, hospital, police, restaurant, taxi, and train. Each dialog consists of a sequence of user utterances and system responses, all annotated with the corresponding dialog state and system action.

We follow the preprocessing procedure from Zhang et al. (2020a) to delexicalize slot values for each system response, and use the standardized evaluation script released by Nekvinda and Dušek (2021), which has also been adopted by the official MultiWoZ dataset.

### 4.2 Evaluation Metrics

In our experiments, we primarily consider the task of end-to-end response generation. In MultiWoZ, response generation performance is evaluated by a combination of three metrics: **Inform rate** measures whether the system has provided an appropriate entity; **Success rate** measures whether the system has answered all the requested attributes; **BLEU** measures the fluency as compared to the references, which are also delexicalized. Finally, the **Combined** score is calculated as $(\text{Inform} + \text{Success}) \times 0.5 + \text{BLEU}$.

### 4.3 Model Architecture and Baseline

In this work, we use GODEL-base (Peng et al., 2022) as a backbone, which is a T5-base model (Raffel et al., 2020) pretrained on both texts and dialog datasets (except MultiWoZ).

**Baseline** We use MTTOD (Lee, 2021), which achieves previous state-of-the-art performance in response generation by performing SL with additional multi-task training. We re-train MTTOD

with GODEL-base (Peng et al., 2022) as the backbone, and report this as *Baseline (MTTOD)*.

**KRLS** Since KRLS targets at improving response generation, we replace the SL objective during response training in MTTOD with the KRLS algorithm, which involves both SL and RL training. We report this result as *KRLS*.

**finetune+KRLS** As the generation procedure in KRLS is based on the assumption stated in Section 3.1, we first initialize the model with an $\mathcal{L}_{\text{SL}}$-finetuned checkpoint, and then perform the same KRLS training procedure as used in *KRLS*. We report this result as *finetune+KRLS*.

More details in training/hyperparameters can be found in Appendix D.

### 4.4 Main Results

Table 1 summarizes the results of end-to-end response generation performance on MultiWoZ. As shown in Table 1, when trained with KRLS directly from backbone (*KRLS* in Table 1) we achieve an improvement of 1.4 in Combined Score compared to the baseline, which mostly comes from increased inform and success rate. Since inform/success rate evaluates how often informable/requestable slot values (i.e. keywords) are generated correctly, this suggests that KRLS can help reinforce a model's ability to generate key tokens (see Section 6.1).

When trained from an $\mathcal{L}_{\text{SL}}$-finetuned checkpoint (*finetune+KRLS*), KRLS further improves to a combined score of 103.8, with major improvements again in the success and inform rate. We believe this is because, as the model has already been finetuned on the entire training dataset, the assumption mentioned in Section 3.1 is better satisfied (see Appendix F for examples). Then, KRLS can better improve a model's keyword generation ability as compared to the case when trained from backbone.

Table 2 compares the training speed of the KRLS algorithm with standard RL training which uses auto-regressive generation. During standard RL training, we removed the SL step in Algorithm 1, and only use a terminal reward during RL as newly generated sequences no longer have a one-to-one mapping to the tokens in gold response (more details in Appendix J). We then measure the total wall-clock time per epoch spent by each algorithm during training and separately during experience collection (line 7-14 in Algorithm 1). While additionally initializing KRLS with a $\mathcal{L}_{\text{SL}}$-finetuned checkpoint (*finetune+KRLS*) achieves a better per-

| Model | Backbone | Response Generation | | | |
|---|---|---|---|---|---|
| | | Inform | Success | BLEU | Combined |
| SOLOIST (Peng et al., 2021) | GPT-2 | 82.3 | 72.4 | 13.6 | 90.9 |
| DoTS (Jeon and Lee, 2021) | BERT-base | 80.4 | 68.7 | 16.8 | 91.4 |
| UBAR (Yang et al., 2021b) | DistilGPT-2 | 83.4 | 70.3 | 17.6 | 94.4 |
| PPTOD (Su et al., 2022) | T5-base | 83.1 | 72.7 | 18.2 | 96.1 |
| BORT (Sun et al., 2022a) | T5-small | 85.5 | 77.4 | 17.9 | 99.4 |
| MTTOD (Lee, 2021) | T5-base | 85.9 | 76.5 | 19.0 | 100.2 |
| GALAXY (He et al., 2022) | UniLM-base | 85.4 | 75.7 | 19.0 | 100.2 |
| Mars-G† (Sun et al., 2022b) | T5-small | 88.9 | 78.0 | **19.9** | 103.4 |
| Baseline (MTTOD) | GODEL-base | 86.0 | 77.4 | 18.9 | 100.6 |
| KRLS | GODEL-base | 87.3 (87.2±0.3) | 78.3 (78.2±0.5) | 19.2 (19.1±0.3) | 102.0 (101.9±0.5) |
| finetune+KRLS | GODEL-base | **89.2** (89.2±0.3) | **80.3** (80.0±0.4) | 19.0 (19.0±0.2) | **103.8** (103.5±0.4) |

Table 1: MultiWoZ 2.2 end-to-end response generation evaluation. Results are "best run $(\mu, \sigma)$" over three runs. The results of previous works are from the official leaderboard of MultiWOZ. † indicates concurrent work.

| Algo | Generation Time | Training Time |
|---|---|---|
| KRLS | 48 min/epoch | 306 min/epoch |
| std. RL | 172 min/epoch | 362 min/epoch |

Table 2: Training speed comparison between KRLS and RL. In standard RL (*std. RL*), auto-regressive sequence generation is used for experience collection.

| Metric | ft+KRLS Win | MTTOD Win | Tie |
|---|---|---|---|
| Fluency | 34.7% | **48.0%** | 17.3% |
| Appropriateness | **55.3%*** | 30.7% | 14.0% |
| Informativeness | **60.7%*** | 26.7% | 12.7% |
| Overall | **59.3%*** | 29.3% | 11.3% |

Table 3: Human evaluation on the generated responses. * indicates $p < 0.01$. Fluency result has no statistical significance due to large variances among annotators.

formance, we note that the same procedure is often used for RL training in NLP (Ramamurthy et al., 2022). Due to the large exploration space for language models, RL algorithms may require many more epochs to train without initializing from a finetuned checkpoint (see Appendix H). Therefore, we focus our comparison solely on running the KRLS algorithm and the standard RL algorithm.

As shown in Table 2, the experience collection time (*Generation Time*) for KRLS is much shorter than RL using auto-regressive generation, as in KRLS only a single forward pass is needed for sequence generation. However, since KRLS additionally includes SL (68 min/epoch) and a per-token reward computation, the total training time per epoch becomes 306 min/epoch, though still 15% faster than the 362 min/epoch with standard RL, which only includes experience collection and PPO training (190 min/epoch).

### 4.5 Human Evaluation

We consider the possibility that automatic metrics from MultiWoZ may not correlate well with human judgements (Liu et al., 2016; Lubis et al., 2022). Thus, we ask crowd-workers on Amazon Mechanical Turk to compare responses generated by baseline (*MTTOD*) and finetune+KRLS (*ft+KRLS*). The

responses are rated in terms of their 1) *appropriateness*[2], 2) *fluency*, 3) *informativeness*, and 4) *overall* quality given a dialogue context. We randomly picked 50 turns in the test set, and provided the generated responses without delexicalization and the dialogue history up to that turn. For each metric, the crowd-workers were to choose which response is better, or if it is a tie. We collected preference from 3 crowd-workers per sampled turn.

Table 3 summarizes the human evaluation results. Our method has been rated more appropriate, informative, and overall more preferred by Turkers. We believe this coincides with the results in Table 1 that our method performs better in inform and success rate, by providing more relevant key information. There is no statistical significance in the fluency result (p > 0.05), which is expected given both models' comparable BLEU scores. Specifically, only 16% of the dialogues have more than one annotator rating MTTOD as more fluent. We believe this is because many pre-trained LMs can already generate fluent texts, and it is often challenging for humans to notice the difference.

---

[2] For *appropriateness* and *fluency*, we followed the definitions from prior work (Zhang et al., 2020b; Ramachandran et al., 2021; Jang et al., 2022; Feng et al., 2023).

| Model | 5% | | | | 10% | | | | 20% | | | |
|---|---|---|---|---|---|---|---|---|---|---|---|---|
| | Inform | Success | BLEU | Combined | Inform | Success | BLEU | Combined | Inform | Success | BLEU | Combined |
| MTTOD | 51.1 | 20.7 | 9.5 | 46.6 | 63.1 | 44.4 | 13.8 | 67.7 | 75.0 | 61.0 | 16.8 | 84.8 |
| ft+KRLS | **55.0** | **22.7** | **11.5** | **50.3** | **64.8** | **47.4** | **15.4** | **71.9** | **78.9** | **65.0** | **17.2** | **89.2** |

Table 4: MultiWoZ end-to-end response generation performance using 5%, 10%, and 20% of training data. "ft+KRLS" refers to *finetune+KRLS*. Both models use GODEL-base as backbone. Results are shown as mean values over three runs.

## 4.6 Low Resource Experiment

Since large and well-annotated dialogue datasets such as MultiWoZ (Budzianowski et al., 2018) is not easy to create in practice, we also investigate KRLS's performance under a low-resource regime. We use 5%, 10%, and 20% of training data to train both baseline (*MTTOD*) and *finetune+KRLS*, and report their performance in Table 4. In Table 4 we find our method outperforms the baseline in all settings. We also find large contributions often from improving inform and success scores, which indicates the effectiveness of KRLS at key token learning without abundant training data.

## 5 Ablation Studies

## 5.1 KRLS Algorithm Ablation

Since KRLS effectively combines SL and RL, we illustrate the contribution of each component in Table 5. During *SL Only* and *RL Only*, we remove the RL training and SL training in KRLS, respectively. In *SL+GOLD*, we replace our RL procedure with GOLD (Pang and He, 2021), which is an offline and off-policy RL algorithm that learns from the gold demonstrations without any generation (see Appendix K for more details), so that we can also isolate the impact of additionally generating a sequence in our approach. Since GOLD uses Policy Gradient (Sutton et al., 1999) without clipping, we do the same with KRLS here (denoted *KRLS(PG)*).

As shown in Table 5, using *SL+GOLD* only achieved a similar performance as compared to *SL Only*. We believe that this is because *GOLD* only intends to learn from the gold responses, while KRLS(PG) also attempts to explore other sequences/tokens to reinforce its keyword learning. Additionally, if the SL objective is removed from KRLS when directly trained from backbone, the performance degrades as the assumption mentioned in Section 3.1 becomes harder to satisfy (c.f. KRLS in Table 1, and discussions in Appendix H).[3]

---

[3]In our prior study, we also experimented with a simple alternative of only using weighted SL on key tokens, remov-

| Algo | Inform | Success | Bleu | Total |
|---|---|---|---|---|
| SL Only | 86.0 | 77.4 | 18.9 | 100.6 |
| RL Only | 84.2 | 72.2 | 17.5 | 95.7 |
| SL+GOLD | 86.1 | 77.1 | 18.8 | 100.4 |
| KRLS(PG) | **88.7** | **78.7** | **19.1** | **102.8** |

Table 5: KRLS Ablation Study. The first two are trained directly from the backbone, and the latter two are trained from a $\mathcal{L}_{SL}$-finetuned checkpoint (see more details in Appendix K).

## 5.2 Reward Function Ablation

In Table 6, we empirically compare our proposed reward with 4 different reward functions and show that: a) providing a per-token reward in addition to providing a terminal reward for the entire sequence is helpful, and b) a fine-grained, context-aware reward that correctly factors in our generation procedure can further improve performance.

In this experiment, we replace our proposed reward in Section 3.2 (denoted as *Prob.* in Table 6) with the following alternatives: *Zero*, which assigns a zero score to all generated tokens, hence only uses the terminal reward for training; *Error*, which assigns a hard penalty of $\pm\mu$ whenever the generated token is correct/incorrect; *BERTS.*, which uses the core mechanism in BERTScore (Zhang et al., 2019) to measure the semantic similarity between the generated tokens and the gold tokens (see Appendix G for more details); *Static.*, which takes the static, context-*unaware* token embeddings from the embedding layer of GODEL-base and compute their cosine similarity as reward. In all cases, the same set of hyperparameters is used to make results more comparable.

As shown in Table 6, our proposed token-level

---

ing RL entirely. However, weighted SL only yields a minor improvement compared to the baselines, reaching a score of 101.2. KRLS using a similar reward function (*Error*, see Section 5.2) already achieves 102.5. We believe this is because KRLS rewards/penalizes generated tokens sampled from the model's distribution using RL, while SL only uses the gold tokens. This finding motivates KRLS to use RL with a reward function emphasizing on keyword learning.

| Finetune+KRLS | | | | |
|---|---|---|---|---|
| Reward | Inform | Success | BLEU | Total |
| *None* | 86.0 | 77.4 | 18.9 | 100.6 |
| Zero | 88.3 | 77.9 | 18.9 | 102.0 |
| Error | 88.8 | 78.5 | 18.8 | 102.5 |
| Static. | 88.7 | 78.4 | 18.8 | 102.4 |
| BERTS. | 88.5 | 78.7 | 18.9 | 102.5 |
| Prob. | **89.2** | **80.3** | **19.0** | **103.8** |

Table 6: KRLS using different per-token reward when trained from a finetuned checkpoint. *None* refers to the baseline of training only with supervised learning.

reward (*Prob.*) outperforms all other alternatives. Interestingly, all reward functions that specified a per-token reward (i.e. *Error*, *BERTS.*, *Static.*, *Prob.*) achieved improvements over *Zero*, which only relies on the terminal reward. This indicates that a more fine-grained per-token reward function is helpful. Additionally, *Prob.* improves upon *Error*, *BERTS.*, and *Static.*, because it additionally factors in our generation procedure that the generated sequence is conditioned on the gold response. Therefore, it can also correctly capture the contextual relationship of the generated tokens.

# 6 Analysis

## 6.1 Keyword Learning

Since KRLS aims to improve the model's ability to generate key information correctly, we track the model's accuracy in generating key tokens during training and validation. In this experiment, we feed in the gold contexts up to the key tokens, and the model is tasked to generate the next token. We then calculate the accuracy by measuring how often the generated token matches the gold key token.

As shown in Figure 4, only performing SL (*baseline*) leads to a slow increase in keyword generation accuracy during early training, as the model focuses on learning other non-key tokens due to their abundance. On the other hand, KRLS periodically uses RL to help the model focus on learning key tokens, which leads to a higher keyword generation accuracy throughout both training and validation (more details in Appendix I).

## 6.2 Error Analysis

Despite reaching a higher inform rate and success rate as more key tokens are generated correctly, we still observe responses that miss some key tokens. We found that these errors often originate

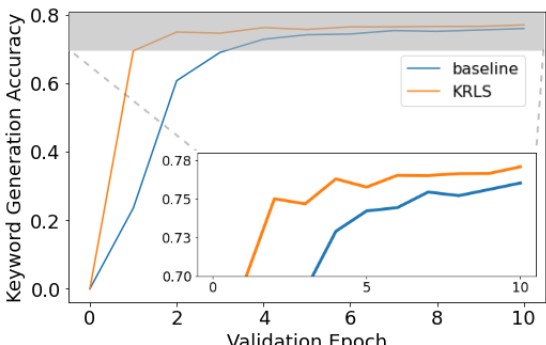

Figure 4: Keyword generation accuracy during validation. *Baseline* is trained with only supervised learning, $\mathcal{L}_{\text{SL}}$. Both models are trained directly from backbone to additionally demonstrate the difference during early training.

| Algo | Inform | Success | BLEU | Total |
|---|---|---|---|---|
| KRLS | 89.2 | 80.3 | 19.0 | 103.8 |
| +DST | 93.1 | 83.7 | 19.1 | 107.5 |
| +Both | 93.5 | 90.9 | 29.8 | 122.0 |
| *Train* | 93.7 | 90.9 | - | - |

Table 7: Test performance of KRLS when generating with Gold Dialog State (+DST), and with both Gold Dialog State and Gold System Act (+Both). *Train* is the performance of the training dataset. Note that +DST is the same as the "policy optimization" task.

from incorrectly generated dialog states and system acts (see Appendix O for examples). This is understandable, as we only used KRLS to improve the response generation component.

To quantify these errors, we additionally use our KRLS-trained model to generate responses when a) the gold dialog state is provided (+*DST*) and b) both the gold dialog state and the gold system action are provided (+*Both*). We present this result in Table 7, and found that +*DST* improved the overall score by nearly 4 points, and +*Both* further improved the overall score by 14.5 points, almost reaching the performance of the training dataset[4]. This shows that much error remains in the DST and system act generation process, so the overall performance can further increase if techniques to separately improve DST and system act generation (e.g., Sun et al. (2022b)) can be combined with KRLS. We leave this for future work.

---

[4]In MultiWoZ, the training dataset includes human errors, hence does not have a perfect inform/success score. Validation/test datasets are hand-picked to only include successful dialogs, so that model performance can be fairly evaluated.

## 7 Related Work

End-to-end dialog systems such as Lei et al. (2018); Yang et al. (2021a); Lee (2021); He et al. (2022) have shown promising results in TOD benchmarks such as MultiWoZ. However, as the standard SL objective does not directly account for TOD metrics such as task success rate, much recent work seeks to incorporate RL techniques to improve TOD performance. In this section, we discuss related applications of RL in TOD, as well as other non-RL-based approaches that have similarities in concept.

**RL for Text Generation** Ranzato et al. (2016); Li et al. (2016); Zhou et al. (2017); Ramamurthy et al. (2022) applies RL to text generation tasks by treating each word as an action and then uses auto-regressive generation to explore high-reward sequences. This results in a large action space for exploration, and these work focuses on methods to stabilize the training process. In principle, these approaches can be modified for TOD tasks, but they would still generally use a user simulator and/or the slow auto-regressive generation step.

**RL for Policy Planning** Many direct applications of RL in TOD focus on optimizing dialog policy planning (Takanobu et al., 2020; Tseng et al., 2021; Lubis et al., 2020). Takanobu et al. (2020); Tseng et al. (2021) jointly optimize both a user system and a dialog system to improve a model's TOD task performance and/or domain adaptation ability, but could be resource intensive as additional user-side training is needed. Alternatively, Lubis et al. (2020); Zhao et al. (2019) uses RL to optimize system action generation in a latent space, but tends to result in the model generating very short responses (i.e., a low BLEU score of 10.8 in MultiWoZ).

**Offline RL in TOD** Many offline RL applications in TOD consider an actor-critic type algorithm (Jang et al., 2022; Verma et al., 2022), which involves using a critic to choose better responses among several generated candidates. These approaches tend to be vulnerable to errors made by the critic model (especially for OOD actions (Levine et al., 2020)), and is resource intensive as multiple auto-regressive generations are needed per episode. Alternatively, Pang and He (2021) proposes the GOLD algorithm, which uses policy-gradient based method in an off-policy setting, by learning solely from the gold demonstrations without any generation/exploration. KRLS additionally performs sequence generations and utilizes gold demonstrations in computing the reward function

(also see Section 5.1 for an empirical comparison).

**Other Notable Related Techniques** Qian et al. (2021) utilizes a student-teacher architecture and MAML (Finn et al., 2017) to improve domain adaptation ability of the student model. Specifically, the teacher model provides weights to scale the NLL loss of each gold token when training the student model. In this aspect, this is similar to GOLD, performing a "weighted learning" on the gold demonstrations. KRLS aims to directly improve TOD performance and achieves this by utilizing RL to perform a "weighted learning" on generated tokens.

## 8 Conclusion

In this work, we explore an approach to utilize RL to improve a model's TOD performance, but also to avoid using a user-simulator or the slow auto-regressive generation process. We propose the Keywords Reinforcement with Next-word Sampling (KRLS) training algorithm, which combines offline RL with a fast sequence generation scheme that directly samples from next-word distributions after supervised training, and a fine-grained per-token reward function that measures the importance and semantic closeness of each generated token. We then evaluate KRLS on the MultiWoZ dataset and show that a) it can help improve E2E response generation performance, reaching new state-of-the-art in the inform rate, success rate, and combined score; b) it can be trained 15% faster than using a standard RL algorithm that performs auto-regressive generation during training/exploration.

## 9 Limitations

Although KRLS is faster to train as it avoids auto-regressive generation, it is difficult for the model to learn/generate sequences significantly different from the gold examples in the dataset. Therefore, this could limit the potential to achieve performance better than the training dataset itself.

Additionally, since during training KRLS creates sequences by conditioning on the gold response, whereas at inference we use auto-regressive generation, the problem of compounding generation error (exposure bias) is re-introduced (Bengio et al., 2015; Venkatraman et al., 2015; Ranzato et al., 2016). Therefore, in this aspect KRLS trades its faster training speed with certain benefits brought by standard RL training in NLP. In the future, it would be worthwhile to explore if a more fine-grained trade-off can be found between an efficient

sequence exploration strategy and those benefits inferred by using auto-regressive generation.

Next, to make KRLS have minimal requirements of extra resources, we avoid using user simulators and perform offline RL training at turn-level. As a result, KRLS does not perform exploration/planning on a dialog-level, which can be very useful for tasks that require long-horizon planning to be successful (e.g., persuading a person to donate to a charity (Wang et al., 2019)). We believe one way to extend KRLS could be using a GPT-like model to learn from an entire dialog, and combine with safe policy improvement methods to avoid potentially large bias and poor sample efficiency during dialog-level RL learning (Ramachandran et al., 2022). We leave this for future work.

Finally, in our runtime experiments (Section 4.4) we found that performing PPO (as well as PG) is a significant bottleneck, taking up more than half of the total training time. Future work may wish to consider ways to improve the speed/memory efficiency[5] of computing those RL objectives to further reduce training time.

## 10 Ethical Considerations

Our work describes an algorithm to improve a model's TOD performance and to expedite the training process. It is aimed at making current TOD systems easier to train, and also better at helping users to achieve their goals.

Generally, while most algorithms are not designed for unethical usage, there is often potential for abuse in their applications. In our experiments, we apply KRLS on the MultiWoZ (Budzianowski et al., 2018) dataset, to improve performance on tasks such as restaurant booking and hotel reservation. However, because TOD training algorithms are typically task-agnostic, it is possible to use them for unethical tasks, such as scamming. We do not condone the use of KRLS for any unlawful or morally unjust purposes.

Additionally, since our experiments use pretrained language models, another concern is on their (in)ability to generate safe, respectful content (Welbl et al., 2021; Gehman et al., 2020). Our work specifically focuses on improving TOD performance, and hence we caution users against any

---

[5]In our implementation, we noticed that certain computations could be cached to save time. However, we found it infeasible in our setting due to limited GPU memory. Future work may also investigate ways to improve memory efficiency in our implementation, to allow for potential speedups.

potential unsafe/toxic/offensive responses generated from the models. Without safety guardrails such as Arora et al. (2022); Lu et al. (2022), we do not advocate using any of our trained models in production settings.

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

## A  KRLS RL Details

To avoid high variance during policy gradient:

$$\nabla \mathcal{L}(\theta) = -G_t \nabla \log p_\theta(x_t|c)$$

we consider a clipped version of this objective, borrowing from proximal policy gradient (PPO) (Schulman et al., 2017) to provide a more stable training and prevent the policy from moving too far away from the pretrained language model. Similar to Wu et al. (2021), we consider optimizing the following surrogate objective:

$$\mathcal{L}_{\text{RL}} = -\min \begin{cases} r(\theta)\hat{A}_t, \\ \text{clip}(r(\theta), 1-\epsilon, 1+\epsilon)\hat{A}_t, \end{cases}$$
(5)

where $\hat{A}_t = -V_\theta(s_t) + \sum_l \gamma^l r_{t+l}$ is the advantage function (Schulman et al., 2015b), $\epsilon$ is the clipping parameter, and $r(\theta)$ is the ratio of the new policy to the old policy $p_\theta^{\text{old}}$:

$$x^{\text{gen}} \sim p_\theta(x|c), \quad r(\theta) = \frac{p_\theta(x^{\text{gen}}|c)}{p_\theta^{\text{old}}(x^{\text{gen}}|c)} \quad (6)$$

In practice, we found that adding additional value function heads $V_\theta$ would 1) increase model size, making it difficult to train under our GPU setting and 2) since KRLS training is performed over a limited number of epochs (Appendix E), we found fitting a small value function head can result in high variance during training. As such, we fixed the value function to be zero and used the return as an estimate $\hat{A}_t \approx \sum_l \gamma^l r_{t+l} = G_t$, which also signals the model the correct tokens to generate. We note that according to Schulman et al. (2015b) page 4, $G_t$ is a $\gamma$-just advantage estimator for $\hat{A}_t$.

For simplicity, we refer "KRLS" to use this clipped policy gradient objective unless explicitly mentioned otherwise.

## B  KRLS using PPO v.s. PG

As shown in Table 8, we found that using the clipped objective (Equation (5)) in KRLS can achieve better performance as compared to simple PG, which can cause high gradient variance (Schulman et al., 2015a; Wu et al., 2021) during training. We believe that this is also due to the clipped objective preventing the new policy $p_\theta$ to move too far away from the old policy, which is useful in our approach as we approximated sequence generation by "sampling" from the gold response. Therefore, we believe that being more pessimistic (Schulman et al., 2017) about each $\theta$ update can be beneficial in our setting.

| | Finetune+KRLS | | | |
| RL Algo | Inform | Success | BLEU | Total |
|---|---|---|---|---|
| PG | 88.7 | 78.7 | **19.1** | 102.8 |
| PPO | **89.2** | **80.3** | 19.0 | **103.8** |

Table 8: Performance comparison when training KRLS with Policy Gradient (PG) and a clipped version (PPO).

## C  Modeling Details

We use a GODEL-base model as backbone, which is an encoder-decoder architecture like T5-base, with ~220M parameters. Similar to MTTOD, an additional decoder is initialized for response generation (the other decoder for DST), which results in an additional ~160M parameters. The encoder is shared for both DST and response generation. This results in a total model size of ~380M parameters.

## D  Implementation and Training Details

For all experiments (including (re)training MT-TOD), we use Adam (Kingma and Ba, 2015) for optimization, a linear schedule with an initial learning rate of $5e^{-5}$ and warm-up steps $= 0.2\times$total training steps.

For *Baseline*, we re-train MTTOD (Lee, 2021) using publicly released code and the best set of hyperparameters reported by the author.

For *KRLS* and *finetune+KRLS*, we pick the best set of hyperparameters (see Appendix E for details) using grid search. As task success and inform rate in MultiWoZ is highly correlated with correctly generating the predefined set of informable/requestable slot values such as "[value_address]" in the response, we use $\mu = 5$ for those key tokens and $\mu = 1$ for others. In addition, we add a terminal reward by measuring the F1-score of generated key tokens compared to the gold key tokens (see Figure 3) to measure overall performance in keywords generation. Note that we did not add a BLEU score for terminal reward, as we found the SL training in KRLS is sufficient.

## E  KRLS Hyperparameters

For the reported results of KRLS in MultiWoZ, we use $k = 3$, $\kappa = 0.5 \times$ total steps per epoch, sampling temperature during generation $\tau = 1.1$, top-p during generation of 0.9, terminal reward scale of 5, learning rate of $5e^{-5}$, learning rate decay of $0.2\times$total steps in training, and batch size of 4. When trained from a $\mathcal{L}_{\text{SL}}$-finetuned checkpoint,

we additionally add a regularization term using KL divergence (against the baseline model) with a weighting of 0.01 to reduce over-optimization on the reward function (Ouyang et al., 2022; Ramamurthy et al., 2022; Jaques et al., 2020). During testing, we used auto-regressive generation with greedy decoding (same as Lee (2021)).

All of our experiments are run on a single GPU, NVIDIA RTX A4000. Running KRLS on a ∼380M encoder-decoder model (see Appendix C) for 4 epochs takes about one day, as it consists of 306 min/epoch for training and 32 min/epoch for validation.

## F   KRLS Sampled Response Examples

We provide example comparisons between our generated response using next-word sampling (*SAMPLED*), and responses produced with auto-regressive generation (*GENERATED*) in Figure 9, Figure 10, and Figure 11. All examples are generated from a $\mathcal{L}_{SL}$-finetuned checkpoint. For short responses, we observe that *GENERATED* are similar to *SAMPLED*. When responses get longer, *SAMPLED* responses become less similar to the *GENERATED* ones.

When the model is not yet $\mathcal{L}_{SL}$-finetuned on the dataset, *SAMPLED* responses tend to repeat keywords and can hardly be interpreted as a sequence. For example, given the input context in Figure 9, a sampled response looks like: "[value_stay] [welcome] [value_stay] [value_arrive] [taxi] [train] [value_stay] [value_stay] [value_stay] [value_stay] [value_stay]".

## G   BERTScore for KRLS

To apply BERTScore in our setting, we first treat our sampled sequence as a standalone generated sequence, and use the cosine similarity between the embedding of each pair of token $x_t^{\text{gen}}, x_t^{\text{gold}}$ after passing through a $\mathcal{L}_{SL}$-finetuned GODEL-base as rewards. Note that as we naturally have a one-to-one mapping between the generated and gold sequences, we can skip the maximal similarity matching step.

However, as shown in both Table 6 and Table 13, *BERTS.* does not perform as well as *Prob.*. This is because BERTScore is designed to measure the semantic similarity between two standalone sentences, while in our setting the generated sequence is conditioned on the gold response. Therefore, in cases when many generated tokens are incorrect,

viewing the sampled sequence as a standalone generated sentence will distort each token's contextual meaning, leading to sub-optimal performance.

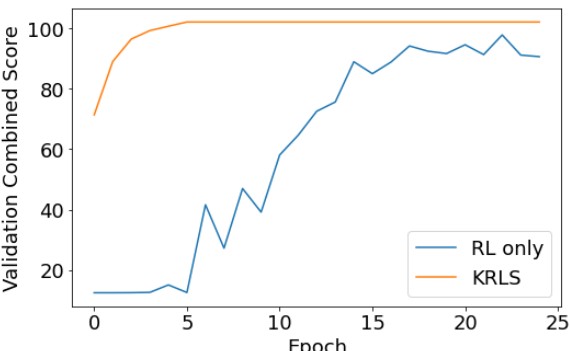

Figure 5: Validation performance of *KRLS* and *RL Only* over time. In *RL Only*, we remove the SL objective from KRLS, and for each sampled episode we additionally append its corresponding gold response.

| Algo | Inform | Success | BLEU | Total |
|---|---|---|---|---|
| SL only | 86.0 | 77.4 | 18.9 | 100.6 |
| RL only | 84.2 | 72.2 | 17.5 | 95.7 |
| KRLS | **87.3** | **78.3** | **19.2** | **102.0** |

Table 9: Performance of individual components of the KRLS Algorithm when trained directly from backbone.

## H   KRLS Directly from Backbone

In Table 9, we present the results when training directly from from the backbone. *SL only* refers to the baseline of only training with SL objective. *RL Only* refers to the KRLS algorithm with SL objective removed. *KRLS* refers to the full KRLS algorithm. When trained directly from backbone, removing the SL objective (*RL Only*) degrades the performance as the assumption mentioned in Section 3.1 becomes harder to satisfy especially during early training (see Figure 5). This is sensible because, without prior SL training, many sequences generated from our sampling method can be highly different from the auto-regressive generated ones. When SL training is included in KRLS, the overall performance improves by 1.4 points, as the additional SL training makes it easier to satisfy our assumption and also made training much smoother (see Figure 5).

## I   Additional Keyword Learning Curves

In addition to the keyword generation accuracy during validation when trained directly from backbone

(see Section 6.1), in this section we also show: a) keyword generation accuracy when trained during both training and validation in Figure 6; b) overall generation accuracy learning curves in Figure 7; c) keyword and overall generation accuracy curves when trained from a $\mathcal{L}_{\text{SL}}$-finetuned checkpoint in Figure 8. Overall generation accuracy is measured by how often a generated token $x_t^{\text{gen}}|x_{1:t-1}^{\text{gold}}, c$ matches the ground truth $x_t^{\text{gold}}$, whether $x_t^{\text{gold}}$ is a key token or a non-key token. Keyword generation accuracy only performs the above comparison when the ground truth token is a key token.

As shown in Figure 6, KRLS can achieve higher keyword generation accuracy than baseline during both training and validation. We believe this is because the RL component in KRLS, especially during the early stages of training, can help the model also learn the less abundant but more important keywords as it has a higher reward. As a result, in Figure 7, KRLS training can also achieve a higher overall generation accuracy.

When trained from a $\mathcal{L}_{\text{SL}}$-finetuned checkpoint as shown in Figure 8, KRLS further increases its keyword generation accuracy. However, in Figure 8(b), the overall generation accuracy is similar to a $\mathcal{L}_{\text{SL}}$-finetuned *baseline*. We believe this is because, after the model has learned to generate most of the tokens correctly, it needs to maintain a balance between over-generating keywords (lower overall accuracy) and correctly generating the keywords (higher keyword generation accuracy).

## J  RL with Auto-Regressive Gen. Setup

In Section 4.4, we compared the training time between normal RL with auto-regressive generation, and our KRLS algorithm. As KRLS has an additional SL step during training, we removed this component to provide a fairer comparison against the normal RL procedure, which usually only includes an auto-regressive sequence generation step during experience collection and PPO training. However, after auto-regressive generation, the generated tokens no longer have a one-to-one mapping to the gold response. Therefore, in this setting we used a zero reward for each token, and a terminal reward of keywords $F_1$ (same as KRLS) as well as a BLEU score (as SL training is removed).

In addition to a faster training speed as shown in Section 4.4, we found that our approach can also reach a better overall performance in Table 10 when trained from a $\mathcal{L}_{\text{SL}}$-finetuned checkpoint for

4 epochs. We believe that this is because, without a fine-grained per-token reward, *RL* might need many more epochs to figure out the importance of those keywords.

| Algo | Inform | Success | BLEU | Total |
|------|--------|---------|------|-------|
| RL | 88.5 | 79.3 | 18.8 | 103.1 |
| KRLS | **89.2** | **80.3** | **19.0** | **103.8** |

Table 10: Test performance when trained from a $\mathcal{L}_{\text{SL}}$-finetuned checkpoint for 4 epochs. *RL* refers to removing the SL step in KRLS, and replacing sequence "sampling" step with auto-regressive generation.

## K  Ablation Study Setup: GOLD

GOLD (Pang and He, 2021) is an offline, off-policy RL algorithm that directly learns from the gold examples in the dataset without any generation step. As the RL component in KRLS is also an offline RL algorithm, we compared KRLS to GOLD in Section 5.1.

To implement GOLD in our experiments, we followed the descriptions in Pang and He (2021), and replaced the RL component in KRLS with GOLD. We denoted this as *SL+GOLD* in Table 5. Additionally, as GOLD uses the simple policy gradient (PG), for a fair comparison we also replaced our PPO objective with PG in KRLS in this experiment. Finally, we kept our per-token reward function $\mathcal{R}$ in *SL+GOLD*, as the reward function proposed by Pang and He (2021) is aimed at optimizing other metrics such as perplexity.

## L  Proof of SL Equivalence

Applying our definition of $\mathcal{R}$ (see Section 3.2) in Equation 3 we get, if $x^{\text{gen}} = x^{\text{gold}}$ *is generated correctly and corresponds to* $\mathcal{R} = 1$, and with the discount factor $\gamma = 0$:

$$\nabla \mathcal{L}(\theta) \propto -\nabla \log p_\theta(x^{\text{gen}}|c)$$

this gives the same gradient as the traditional supervised learning for SL in Equation 1.

## M  Effect of $\kappa, \mu$ and $k$ in KRLS

We empirically tested a range of hyperparameters for KRLS, including $\kappa \in \{0.1, 0.5, 1.0\} \times$ steps per epoch, $\mu \in \{2, 5, 10\}$, and $k \in \{1, 3, 5\}$. We present the results in Table 11 and Table 12.

| Finetune+KRLS | | |
| | $\mu = 2.0$ | $\mu = 5.0$ | $\mu = 10.0$ |
| --- | --- | --- | --- |
| $\kappa = 0.1n$ | 102.2 | 102.4 | 102.3 |
| $\kappa = 0.5n$ | 103.3 | 103.8 | 103.8 |
| $\kappa = 1.0n$ | 102.3 | 102.1 | 102.5 |

Table 11: Effect of different $\kappa$ and $\mu$ on the combined score in MultiWOZ. $n$ represents the number of training steps in an epoch.

| Finetune+KRLS | | |
| | $k = 1$ | $k = 3$ | $k = 5$ |
| --- | --- | --- | --- |
| $\kappa = 0.1n,$ $\mu = 5.0$ | 102.9 | 103.8 | 102.8 |

Table 12: Effect of different $k$ on the combined score in MultiWOZ. $n$ represents the number of training steps in an epoch.

## N Additional Reward Function Ablation

We additionally show the effect of several per-token reward functions in our KRLS algorithm when trained directly from backbone (hence the assumption mentioned in Section 3.1 is harder to satisfy). As shown in Table 13, all variants using KRLS still achieved improvement from baseline (also see Figure 6 and Figure 7). Specifically, *Zero* reward and *Prob.* reward achieved the highest and second highest, with 102.2 and 102.0 as Combined Score, respectively.

| | KRLS | | |
| Reward | Inform | Success | BLEU | Total |
| --- | --- | --- | --- | --- |
| *None* | 86.0 | 77.4 | 18.9 | 100.6 |
| Zero | 87.7 | 78.6 | 19.0 | 102.2 |
| Error | 86.2 | 78.6 | 19.2 | 101.6 |
| BERTS. | 87.2 | 78.1 | 19.0 | 101.7 |
| Static. | 87.2 | 77.6 | 19.3 | 101.7 |
| Prob. | 87.3 | 78.3 | 19.2 | 102.0 |

Table 13: Performance of Training Directly from Backbone with KRLS using Different per-token Reward

## O KRLS Error Examples

We present three examples in Figure 12, Figure 13, and Figure 14 when KRLS trained model does not generate the required key tokens. We observe that in most cases, error originates from incorrectly generated system action and dialog state. This hints

at a direction for further improvement in lines of making dialog state and system action generation more robust (Sun et al., 2022b).

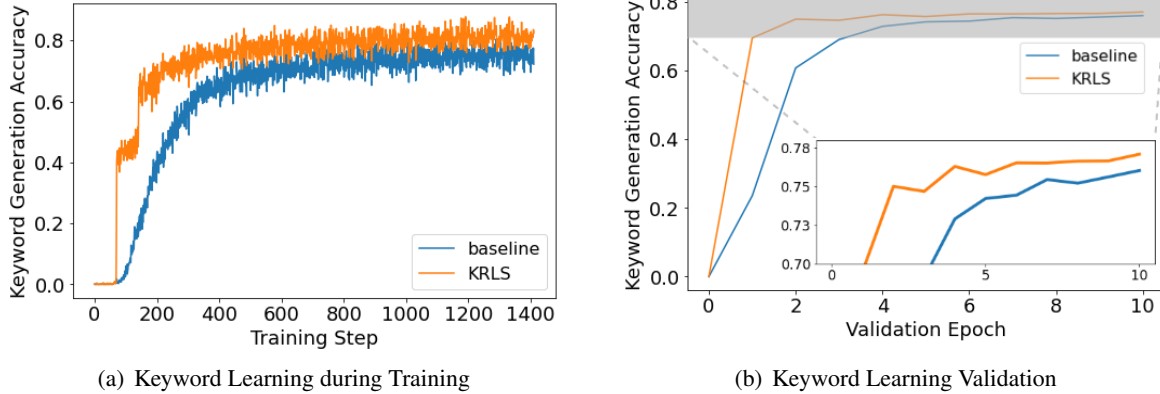

(a) Keyword Learning during Training  (b) Keyword Learning Validation

Figure 6: Keyword Generation Accuracy during Training and Validation. *Baseline* is the standard SL training using $\mathcal{L}_{\text{SL}}$. Both baseline and KRLS are directly trained from backbone.

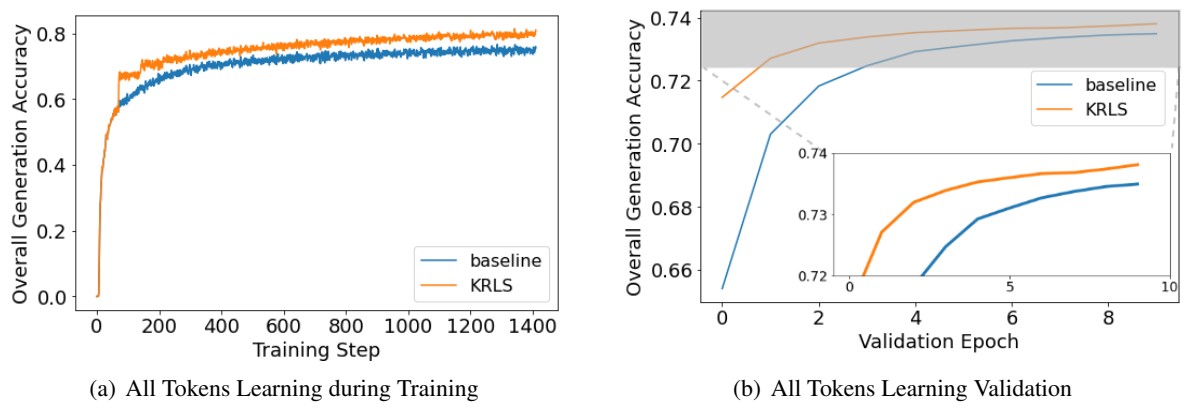

(a) All Tokens Learning during Training  (b) All Tokens Learning Validation

Figure 7: All Token Generation Accuracy during Training and Validation. *Baseline* is the standard SL training using $\mathcal{L}_{\text{SL}}$. Both baseline and KRLS are directly trained from backbone.

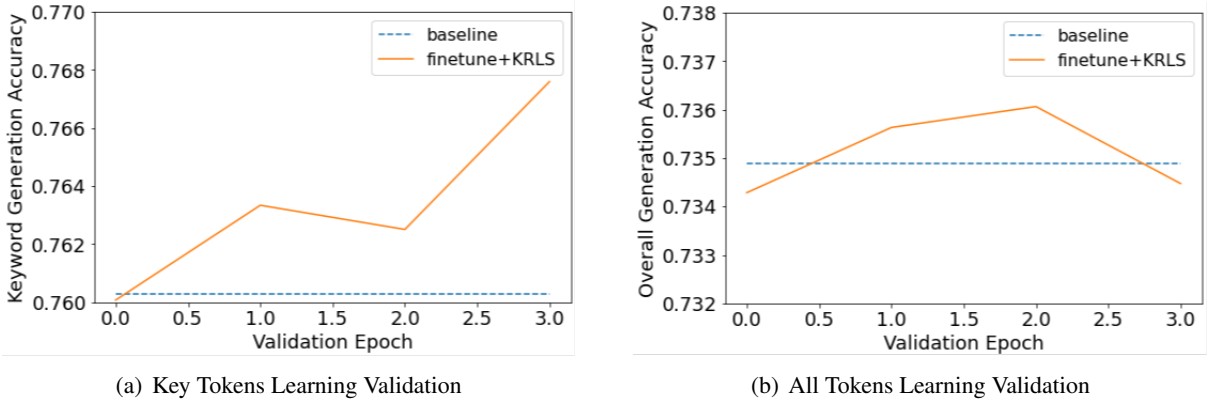

(a) Key Tokens Learning Validation  (b) All Tokens Learning Validation

Figure 8: Key and All Token Generation Accuracy during Training and Validation. *Baseline* is the standard SL training using $\mathcal{L}_{\text{SL}}$. *finetunt+KRLS* is trained from a $\mathcal{L}_{\text{SL}}$-finetuned checkpoint, i.e. *Baseline*.

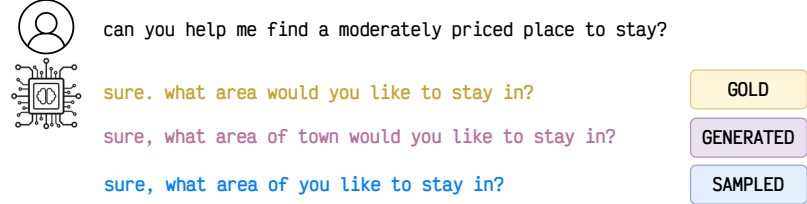

Figure 9: Example sequence generated by KRLS. Texts in black are the user's utterances. *GOLD* represents the gold response. *GENERATED* represents response produced using auto-regressive generation. *SAMPLED* represents response produced using the next-word sampling method in KRLS.

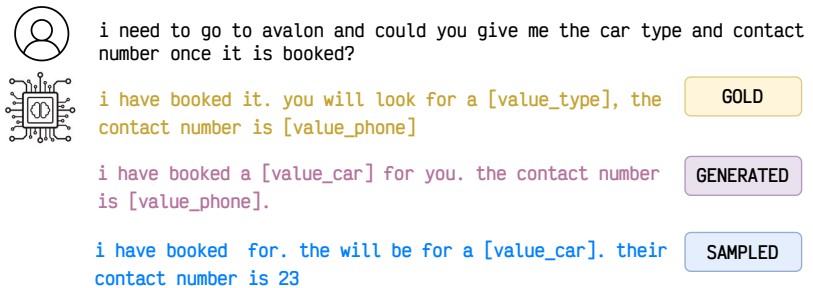

Figure 10: Example sequence generated by KRLS. Texts in black are the user's utterances. *GOLD* represents the gold response. *GENERATED* represents response produced using auto-regressive generation. *SAMPLED* represents response produced using the next-word sampling method in KRLS.

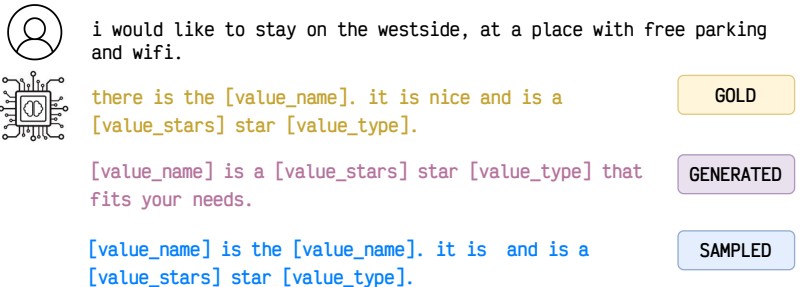

Figure 11: Example sequence generated by KRLS. Texts in black are the user's utterances. *GOLD* represents the gold response. *GENERATED* represents response produced using auto-regressive generation. *SAMPLED* represents response produced using the next-word sampling method in KRLS.

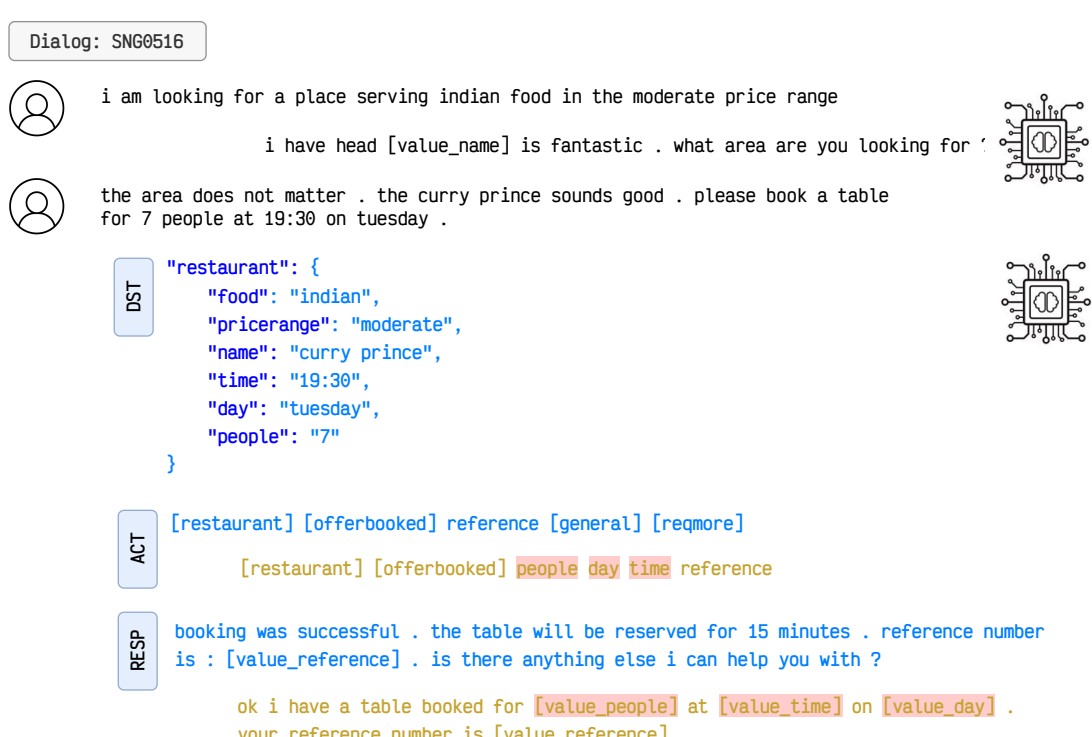

**Dialog: SNG01380**

i am looking for a particular restaurant . its name is called restaurant alimentum

**DST**
```
"restaurant": {
    "name": "restaurant alimentum"
}
```

**ACT**
[restaurant] [inform] food price area name [offerbook]

[restaurant] [inform] food price area address name postcode phone

**RESP**
[value_name] is an [value_pricerange] [value_food] restaurant in the [value_area] . would you like me to make a reservation for you ?

[value_name] offers [value_food] found [value_area] has [value_pricerange] price range [value_phone] [value_postcode] [value_address]

Figure 12: KRLS Generation Error Example 1. Texts in black are the input context, in blue are the generated tokens, and in yellow/gold are the ground truth. Texts highlighted in red are incorrect/missing key tokens compared to the ground truth.

**Dialog: SNG0516**

i am looking for a place serving indian food in the moderate price range

i have head [value_name] is fantastic . what area are you looking for '

the area does not matter . the curry prince sounds good . please book a table for 7 people at 19:30 on tuesday .

**DST**
```
"restaurant": {
    "food": "indian",
    "pricerange": "moderate",
    "name": "curry prince",
    "time": "19:30",
    "day": "tuesday",
    "people": "7"
}
```

**ACT**
[restaurant] [offerbooked] reference [general] [reqmore]

[restaurant] [offerbooked] people day time reference

**RESP**
booking was successful . the table will be reserved for 15 minutes . reference number is : [value_reference] . is there anything else i can help you with ?

ok i have a table booked for [value_people] at [value_time] on [value_day] . your reference number is [value_reference] .

Figure 13: KRLS Generation Error Example 2. Texts in black are the input context, in blue are the generated tokens, and in yellow/gold are the ground truth. Texts highlighted in red are incorrect/missing key tokens compared to the ground truth.

Dialog: MUL2177

i am looking for a place in cambridge . it does n't need to include internet and should be a hotel .

i was not able to find a [value_type] type with no internet , but there are [value_type] with internet .

okay , that would be fine , as long as it has free parking .

okay , do you have any preference on price range ?

no . how many are available ?

there are [value_choice] different hotels around the city with free parking . do you have a preference for city area or star rating ?

the area does not matter . the curry prince sounds good . please book a table for 7 people at 19:30 on tuesday .

**DST**
```
"hotel": {
    "type": "hotel",
    "parking": "yes",
    "internet": "no"
    "pricerange": "dont care",
    "area": "dont care",
    "stars": "dont care"
}
```

**ACT**
[hotel] [recommend] name

[hotel] [recommend] name area

**RESP**
i would suggest [value_name] .

it 's hard to say . how about the [value_name] in the [value_area] part of town ?

Figure 14: KRLS Generation Error Example 3. Texts in black are the input context, in blue are the generated tokens, and in yellow/gold are the ground truth. Texts highlighted in red are incorrect/missing key tokens compared to the ground truth.