# OpenReview forum: "KRLS: Improving End-to-End Response Generation in Task Oriented Dialog with Reinforced Keywords Learning"
_EMNLP/2023/Conference — EMNLP 2023 Main_

### Official Review · Reviewer_uymB · 2023-08-02

**Soundness:** 4

**Excitement:**

4: Strong: This paper deepens the understanding of some phenomenon or lowers the barriers to an existing research direction.

**Paper Topic And Main Contributions:**

This paper proposes KRLS, an RL-based method to reinforce model generate responses with keywords. To reinforce keywords generation in from models, authors design a token-wised reward function by using a probability from a supervised fine-tuned decoder model given golden input, then scale the reward for keyword to ensure the keywords have higher probabilities to be generated.
In addition, since the typical RL methods relied on autoregressive generation process which takes time to sample trajectories, this paper introduces using next word probability with a supervised LM to speed up the sample process.
In their experiments, they used MTTOD as a supervised model before apply RL.
The authors evaluate the performance on well-known TOD metrics that includes BLEU, inform, success, and bleu. They also compare the several state of the art baselines and conduct the human evaluation.
The experimental results show that KRLS performs better than baseline on the metrics, the generation process is also faster than standard RL.

**Questions For The Authors:**

Since you have already has these kind of 'annotated' data from supervised model with weights, have you tried use normal supervise learning such as weighted cross entropy to finetune the model?

In next word sampling process, because each sampled token was based on golden response and it was not based on previous sample results, can I say the length of the trajectories is equal to 1?

**Reasons To Accept:**

This paper is well-written and easy to follow.
The proposed method is very straightforward and particle, the authors understand the importance of keywords in response for TOD and try to design a reward function with 'weighted' (importance) probabilities from another decoder models.
The experiments results are solid, the compared baselines, metrics, and ablation study makes sense to me. They also conduct human trials to valid the performance.

**Reasons To Reject:**

None

**Reproducibility:**

5: Could easily reproduce the results.

**Reviewer Confidence:**

4: Quite sure. I tried to check the important points carefully. It's unlikely, though conceivable, that I missed something that should affect my ratings.

---

> ### Author Rebuttal · Authors · 2023-08-29
>
> Thank you for your feedback and acknowledgement of our approach/experiments/analysis!
>
> Response to Questions:
> 1. “have you tried using weighted supervised learning”?
>     \
>     \
>     Yes! In our prior experiments, we performed a weighted supervised learning on key tokens and found a minor improvement of 0.6 points to a score of 101.2 compared to the baseline. KRLS using a similar reward function (see reward named Error in Section 5.2 L419-421) achieves 102.5. We believe this is because KRLS rewards/penalizes *generated* tokens sampled from the model's distribution, while SL only uses the *gold* tokens. However, this prior experiment did motivate our design of reward function emphasizing on keyword learning. We will add this discussion in our final manuscript.
>
> 2. “Can I say the length of the trajectories is equal to 1” in the next word sampling process?
>     \
>     \
>     Yes. Given a gold sequence of length T, next-word sampling samples T tokens in parallel to construct a “generated” sequence. Thus, sampling each of the T tokens using this procedure can be seen as generating a trajectory of length 1 as you described.

---

### Official Review · Reviewer_vXVC · 2023-08-03

**Typos Grammar Style And Presentation Improvements:** Please refer to the above “Paper Topi…
**Soundness:** 3

**Excitement:**

4: Strong: This paper deepens the understanding of some phenomenon or lowers the barriers to an existing research direction.

**Paper Topic And Main Contributions:**

This paper presents an offline RL-based algorithm to improve response generation for Task-Oriented Dialogue (TOD) systems. It employs a fine-grained reward function to guide the agent in focusing on key information, such as slot/slot-value-related words.

I agree with the motivation of having the dialog agent focus on key information, and the use of a fine-grained approach is consistent with the analysis. However, there are several concerns:

1、The core contribution of this paper seems to be the design of a fine-grained reward model based on keywords. However, these keywords need to be manually defined, which can be cumbersome. Additionally, the explanation of the Reward Model section is unclear, and I do not understand how R is calculated when xgen != xgold. The reward definition in the paper is not very convincing and lacks empirical analysis.

2、The next word sampling section merely uses all gold tokens as input and leverages the next word distribution corresponding to each token. This capability is inherent to transformers and does not seem special.

3、The introduction of the baselines, KRLS, and finetune KRLS, is very confusing. Lines 197-198 mention that KRLS replaces SL with the proposed KRLS algorithm, so does KRLS not involve RL? What is the difference between the SL part of finetune+KRLS and KRLS?

Besides, several parts of the paper are hard to follow, such as:

1、Lines 156-164 are confusing. Why can the “assumption” lead to the approximation of “sequence generation by sampling from the next-word distributions conditioned on the gold response? ”

2、Equation 4 does not clarify how R is calculated when xgen != xgold. Moreover, the definitions of xt and xgold are unclear, and they often seem to mean the same thing.

3、In Table 1, the methods in the leftmost column are not defined. What does “normal token” mean in Figure 2?

4、What is the purpose of placing MDP in line 162?

5、The quotation marks in line 298 are inconsistent with the rest of the paper.

Considering the issues mentioned above, I would recommend rejecting this paper.

**Questions For The Authors:**

Please refer to the above “Paper Topic And Main Contributions” part.

**Reasons To Accept:**

Please refer to the above “Paper Topic And Main Contributions” part.

**Reasons To Reject:**

Please refer to the above “Paper Topic And Main Contributions” part.

**Reproducibility:**

3: Could reproduce the results with some difficulty. The settings of parameters are underspecified or subjectively determined; the training/evaluation data are not widely available.

**Reviewer Confidence:**

5: Positive that my evaluation is correct. I read the paper very carefully and I am very familiar with related work.

---

> ### Author Rebuttal · Authors · 2023-08-29
>
> Thank you for your feedback! We will update our final manuscript according to your questions/concerns to improve our clarity!
> \
> \
> Our main contributions are an efficient offline RL algorithm that 1) approximates auto-regressive generation with next-word sampling + SL, and 2) a per-token reward to improve a model’s keyword generation ability. We employed KRLS on MultiWoZ, and show that it can achieve both 15% training time reduction and improvements in test performance. We also presented various ablation studies (Section 5) and analysis (Section 6) to support our results.
>
> 1. a. “The design of a fine-grained reward model is based on keywords, and defining keywords can be cumbersome”
>     \
>     \
>     We focused our study on task-oriented dialogues (as in dedicated in our title) because keywords in those tasks often come readily defined: e.g. restaurant name/address from restaurant booking tasks in the Multiwoz dataset, and movie name/ratings from movie recommendation datasets (L247-252). Additionally, we believe for other dialogue tasks such as QA/social chat, keywords can often be automatically processed and defined, such as the entities mentioned in WikiQA answers (Yang et al., EMNLP 2015), and the intent keywords in ESConv (Liu et al., ACL-IJCNLP 2021).
>     \
>     \
>     b. “reward definition in the paper is not very convincing and lacks empirical analysis”
>     \
>     \
>     In Section 5.2, we empirically compared our reward function with 4 other common reward functions. In Table 5, we find that our reward function (denoted as Prob., L415-416) performed the best (103.8 points), compared against alternatives such as only using terminal reward (denoted as Zero, 102.0 points), and using both terminal reward and BERTScore for a soft penalty on each generated token (denoted BERTS.,102.5 points). Comparisons between all reward functions in Table 5 also showed that a) using a per-token reward is beneficial and b) a context-aware reward can further improve performance (see Section 5.2, L408-414). Both of these aspects are included in our reward function design to achieve the best performance.
>     \
>     \
>     c. “explanation of the Reward Model is unclear”
>     \
>     \
>     We apologize. In sum, Section 3.2 and Equation 4 describes how our reward function is calculated for every token. Our per-token reward function combines two scores, the semantic closeness score of each generated token and its importance score. In general, the closeness score for each token is computed using a decoder network (L200-207), and the importance score $\mu$ is a hyperparameter depending on whether the token is a keyword or not (L212-214). We mainly used $\mu=5$ for key tokens and $\mu=1$ otherwise (for experiments on hyperparameters see Appendix N).
>     \
>     \
>     d. “I do not understand how R is calculated when xgen != xgold”
>     \
>     \
>     In general, R is calculated using Equation 4. Under the special circumstance that a) a generated token is correct, the closeness score is fixed to 1.0, and b) a *keyword* is generated incorrectly, the closeness score is fixed to -1.0 to emphasize keyword learning (L207-211). In all other cases we use the closeness score computed by the decoder (see L200-207), i.e. the general procedure also described in our response above. We will make further clarification on this in our manuscript.
>
> 2. “The next word sampling procedure does not seem special, as it uses the inherent capability of transformers”
>     \
>     \
>     The next-word sampling procedure (Section 3.1) is used by KRLS to approximate and replace the slow auto-regressive generation step used in traditional RL. In traditional RL algorithms, this auto-regressive process is used but is time-consuming as each token in a sequence is decoded **sequentially**. In KRLS, we introduce the next-word sampling procedure which effectively decodes each token **in parallel** by conditioning on the gold context, achieving significant speed improvement (Table 2, and L339-349). When performed after SL steps in KRLS, we find sequences decoded using next-word sampling can then be used as the exploration step (see Q1 from reviewer GMA2 for more details), replacing the traditional auto-regressive generation procedure.
>
> 3. “Introduction of the baselines, KRLS, and finetune KRLS, is very confusing. Does KRLS not involve RL? What’s the difference between finetune+KRLS and KRLS?”
>     \
>     \
>     We are sorry for the confusion. In our main experiments (Section 4.3), the baseline is MTTOD which mainly performs SL. *KRLS* then replaces such SL training with the KRLS algorithm (L296-298), which involves RL. *finetune+KRLS* has identical training procedure as *KRLS*, but is initialized with a SL-finetuned checkpoint (L299-303). Performing RL from a SL-finetuned checkpoint is a commonly used technique for applications in NLP, see Ramamurthy et al, ICLR 2023 for example.
>
> 4. “Why can the assumption (Section 3.1, L159-164) lead to next-word sampling approximating sequences generated by auto-regressive generation?”
>     \
>     \
>     Auto-regressive generates each token sequentially by conditioning on previously generated tokens. Next-word sampling generates each token in parallel by conditioning on the gold context. If we assume that the model “generates sequences similar to the gold responses in the training set” (L160-161), then previously generated tokens are similar to the gold context. This leads to sequences generated by next-word sampling approximating that generated by auto-regressive generation.
>
> 5. a. “Equation 4 does not clarify how R is calculated when xgen != xgold”
>     \
>     \
>     Please see response 1.c and 1.d.
>     \
>     \
>     b. “definitions of xt and xgold are unclear, and they often seem to mean the same thing”
>     \
>     \
>     In L210-211, $x_t$ should be $x_t^{gold}$. We apologize for this typo, and thank you for pointing it out. In all other places, we believe we used superscripts such as $x_t^{gen}$ or $x_t^{gold}$ to differentiate when it is ambiguous. We will update this in our final manuscript.
>
> 6. “In Table 1, methods in the leftmost column are not defined”
>     \
>     \
>     The first 7 methods (up to GALAXY) are official results reported in the MultiWoZ leaderboard as of submission. Mars-G relies on improving the DST/action generation module prior to response generation (L481-486). It is a concurrent work with results published recently but has no public code/model weights. The baseline and our methods (KRLS) in Table 1 are explained in Section 4.3 and our response to question 3. We will add short descriptions of those methods in our final manuscript for clarity.
>
> 7. “What does ‘normal token’ mean in Figure 2?”
>     \
>     \
>     We refer to all non-key tokens as normal tokens. We will clarify this in our final manuscript.
>
> 8. “The quotation marks used in line 298 are inconsistent with the rest of the paper”
>     \
>     \
>     Thank you for pointing this out! We will correct L296, L298, and L303 to use italics.
> ---
> [WikiQA: A Challenge Dataset for Open-Domain Question Answering](https://aclanthology.org/D15-1237) (Yang et al., EMNLP 2015)
> \
> [Towards Emotional Support Dialog Systems](https://aclanthology.org/2021.acl-long.269) (Liu et al., ACL-IJCNLP 2021)
> \
> [Is Reinforcement Learning (Not) for Natural Language Processing: Benchmarks, Baselines, and Building Blocks for Natural Language Policy Optimization](https://arxiv.org/abs/2210.01241) (Ramamurthy et al, ICLR 2023)

---

### Official Review · Reviewer_GMA2 · 2023-08-04

**Soundness:** 4

**Excitement:**

4: Strong: This paper deepens the understanding of some phenomenon or lowers the barriers to an existing research direction.

**Paper Topic And Main Contributions:**

This work proposed a keywords reinforcement Learning framework with next-word sampling.  This method introduces a fine-grained reward function to learn more information about the keywords in a dialog, and reduces the training time compared with the standard Reinforcement Learning method.

**Questions For The Authors:**

1. In section 3.3, training the model with SL over several batches is enough for the assumption in section 3.1? Because the dialogue generation task is challenge. I think maybe the model can't generate a response that reflects the keywords yet.

**Reasons To Accept:**

1. The motivation is clear and reasonable.

2. The paper makes sufficient experiments and analyses to illustrate the effectiveness of the method.

3. The paper is well-organized and clearly writing.

**Reasons To Reject:**

1. Even though the paper demonstrates the improved effectiveness and faster training speed of this method, I think the authors should have explained more about the necessity of this method, considering the close final results of KRLS and baseline in Figure 4, and the fact that KRLS affects the Fluency of the generated responses (human evaluation). I agree that focusing on learning keywords will make it easier for the model to learn the message of the response, but the fluency of the response is still an important evaluation metric.

2. The experimental results show that "finetune + KRLS" achieves significant results, which means that the effectiveness of this method is greatly influenced by the generative model. So I think the comparison in Table 2 is not fair enough. Since the comparison in Table 10 uses "finetune + KRLS", I think the comparison in Table 2 should take into account the time needed to fine-tune the model on the dataset first.

**Reproducibility:**

4: Could mostly reproduce the results, but there may be some variation because of sample variance or minor variations in their interpretation of the protocol or method.

**Reviewer Confidence:**

3: Pretty sure, but there's a chance I missed something. Although I have a good feel for this area in general, I did not carefully check the paper's details, e.g., the math, experimental design, or novelty.

---

> ### Author Rebuttal · Authors · 2023-08-29
>
> Thank you for your feedback and acknowledgement of our comprehensive experiments and analysis!
>
> 1. a. “The final results of KRLS and baseline are close.”
>      \
>      \
>     We agree, but we believe improvements on the MultiWoZ dataset is very difficult. KRLS achieved more than 3 points of improvement compared to a strong baseline (MTTOD), reaching SOTA at a score of 103.8. Mars-G reaching 103.4 is a concurrent work with no public code or model weights available as of 08/24/2023. Additionally, we believe our approach is parallel to Mars-G, which focuses on improving DST and system act generation, while KRLS focuses on improving response generation.
>      \
>      \
>     b. “KRLS affects the Fluency of the generated responses (human evaluation)”
>      \
>      \
>      We note that in Table 3, human evaluation on fluency had **no statistical significance**. Specifically, only 16% of the dialogues has more than 1 annotator rating the MTTOD (baseline) as more fluent. We believe this is because many pretrained-LMs can already generate fluent texts, and it is often hard for humans to notice the difference. Additionally, in Table 1 we find both baseline and KRLS achieving a similar BLEU score of 19.0. We thus believe our method has no significant effect on the fluency of the generated responses. We will clarify this in our final manuscript!
>
> 2. “the comparison in Table 2 is not fair enough” and “should take into account the time needed to fine-tune the model”
>      \
>      \
>      RL training in NLP often **also** initializes with a fine-tuned checkpoint (Ramamurthy et al, ICLR 2023). This is because actions in NLP tasks are often defined by the sequence of tokens in a response/text, hence it is difficult to train from scratch using RL as the exploration space is huge. We also showed this difficulty empirically in Appendix I and Figure 5: training with RL required 25 epochs to achieve a similar task performance compared to KRLS (w/o finetune) after only 5 epochs.
>      \
>      \
>     **However, we agree that this is a good point and should be mentioned in the main text**. We will add clarifications on this in our final manuscript.
> ----
> #### Response to Questions:
> 1. “Is training the model with SL over several batches enough for the assumption in section 3.1?”
>      \
>      \
>     KRLS alternates between SL and RL every epoch to help alleviate the assumption in section 3.1. In our experiments, we found that KRLS improved 1.4 points over baseline to 102.0, while finetune+KRLS further improved to a total score of 103.8. We believe initializing KRLS from a fine-tuned checkpoint helped to better satisfy the assumption and hence further improved the result, as the model has already been trained on the gold utterances several times (L318-237, Section 4.4).
> ----
> [Is Reinforcement Learning (Not) for Natural Language Processing: Benchmarks, Baselines, and Building Blocks for Natural Language Policy Optimization](https://arxiv.org/abs/2210.01241) (Ramamurthy et al, ICLR 2023)

---

### Meta-Review · Area_Chair_f7Jt · 2023-09-18

**Recommendation:** 4

**Metareview:**

Based on the three reviews, the majority of reviewers are positive about the paper's contributions and presentation. Reviewers appreciate the clear motivation and well-organized structure of the paper. The experimental results are considered solid, and the proposed method shows improvement over baselines. The use of a fine-grained reward function to focus on keywords is seen as a valuable contribution.

However, there are also some concerns raised by the reviewers. One reviewer questions the necessity of the proposed method, as the improvement over baselines is not substantial and the fluency of the generated responses is affected. Another reviewer points out the unclear explanation of the reward model and questions the uniqueness of the next word sampling approach. The confusion around the baselines and the comparison in Table 2 is also mentioned.

Overall, the paper is well-written and the proposed method is straightforward and valuable. The experimental results and analysis support the claims. However, there are some areas that need improvement, such as further explanation of the reward model and clearer comparison with baselines. The necessity and effectiveness of the proposed method should be addressed more thoroughly.

---

### Decision · Program_Chairs · 2023-10-07

**Decision:**

Accept-Main

**Comment:**

Based on the three reviews, the majority of reviewers are positive about the paper's contributions and presentation. Reviewers appreciate the clear motivation and well-organized structure of the paper. The experimental results are considered solid, and the proposed method shows improvement over baselines. The use of a fine-grained reward function to focus on keywords is seen as a valuable contribution.

However, there are also some concerns raised by the reviewers. One reviewer questions the necessity of the proposed method, as the improvement over baselines is not substantial and the fluency of the generated responses is affected. Another reviewer points out the unclear explanation of the reward model and questions the uniqueness of the next word sampling approach. The confusion around the baselines and the comparison in Table 2 is also mentioned.

Overall, the paper is well-written and the proposed method is straightforward and valuable. The experimental results and analysis support the claims. However, there are some areas that need improvement, such as further explanation of the reward model and clearer comparison with baselines. The necessity and effectiveness of the proposed method should be addressed more thoroughly.